Genome-wide analysis of the CalS gene family in cotton reveals their potential roles in fiber development and responses to stress

Feng Jiajia 1 2
Chen Yi 1
Xiao Xianghui 2
Qu Yunfang 1
Li Pengtao 2
Lu Quanwei daweianyang@163.com 1 2
Huang Jinling huangjlsxau@163.com 1
1 College of Agriculture, Shanxi Agricultural University , Taigu , Shanxi , China
2 School of Biotechnology and Food Engineering, Anyang Institute of Technology , Anyang , Henan , China
Liu Xin
Electronic publication date: 2021 Nov 30
Publication date: 2021
Volume: 9
Electronic Location ID: e12557
Received 2021 Aug 27; Accepted 2021 Nov 5
Copyright: ©2021 Feng et al.
Copyright year: 2021
Copyright holder: Feng et al.
License: This is an open access article distributed under the terms of the Creative Commons Attribution License, which permits unrestricted use, distribution, reproduction and adaptation in any medium and for any purpose provided that it is properly attributed. For attribution, the original author(s), title, publication source (PeerJ) and either DOI or URL of the article must be cited.
License URL: https://creativecommons.org/licenses/by/4.0/

Keywords: Callose synthase, Synteny, Gene family, Gossypium, Various stresses

Funding: The National Natural Science Foundation of China U1804103 The State Key Laboratory of Cotton Biology Open Fund CB2020A10 This work was supported by the National Natural Science Foundation of China (U1804103), the State Key Laboratory of Cotton Biology Open Fund (CB2020A10). There was no additional external funding received for this study. The funders had no role in study design, data collection and analysis, decision to publish, or preparation of the manuscript.

==============================
Callose deposition occurs during plant growth and development, as well as when plants are under biotic and abiotic stress. Callose synthase is a key enzyme for the synthesis of callose. In this study, 27, 28, 16, and 15 callose synthase family members were identified in Gossypium hirsutum, Gossypium barbadense, Gossypium raimondii, and Gossypium arboreum using the sequence of Arabidopsis callose synthase. The CalSs were divided into five groups by phylogenetic, gene structure, and conservative motif analysis. The conserved motifs and gene structures of CalSs in each group were highly similar. Based on the analysis of cis-acting elements, it is inferred that GhCalSs were regulated by abiotic stress. WGD/Segmental duplication promoted the amplification of the CalS gene in cotton, and purification selection had an important function in the CalS family. The transcriptome data and qRT-PCR under cold, heat, salt, and PEG treatments showed that GhCalSs were involved in abiotic stress. The expression patterns of GhCalSs were different in various tissues. We predicted that GhCalS4, which was highly expressed in fibers, had an important effect on fiber elongation. Hence, these results help us understand the role of GhCalSs in fiber development and stress response.

Introduction

Callose is a linear homopolymer composed of β-1, 3-linked glucose, widely found in higher plants as an important part of specialized cell walls or cell wall-associated structures (Chen & Kim, 2009). However, Callose synthesis mainly depends on callose synthase (CalS) (Hong, Delauney & Verma, 2001). In Arabidopsis thaliana, callose activity was significantly increased and deposited after overexpression of the AtCalS12 (Ellinger et al., 2013). In rice, callose deposition in the plasmodesmata of crr1 (AtCalS10 homologous gene) mutants was reduced (Song et al., 2016).

Callose can regulate the transport of plasmodesmata and phloem, affect plant development and response to biotic stress by controlling callose synthases. For example, the AtCalS5 could maintain the normal formation of callose walls during pollen development (Shi et al., 2015a; Shi et al., 2015b). Meanwhile, CalS was regulated by various signaling pathways, and different biological regulatory processes involve hormones, transcription factors. For instance, external application of ABA can increase the activity of callose synthase of rice, promoting callose deposition, and thus increasing its resistance to BPH insects (Liu et al., 2017). The expression of CalSs were also affected by developmental and stress conditions, such as pollen development (Toller et al., 2008; Xie, Wang & Hong, 2010; Huang et al., 2009; Shi et al., 2015a; Shi et al., 2015b), low-temperature stimulation (Fromm et al., 2013), mechanical wounding (Cui & Lee, 2016; Jacobs et al., 2003; Xie et al., 2011), fungal diseases (Dong et al., 2008; Oide et al., 2013; Ellinger et al., 2014; Blümke, Somerville & Voigt, 2013), bacterial diseases (Granato et al., 2019; Enrique et al., 2011), and insect diseases (Ahmad et al., 2011; Koh et al., 2012; Sun et al., 2018; Yao et al., 2020).

Due to the importance of callose, the callose synthase gene family has been identified in various plants. To date, it has been reported as 12 AtCalSs in Arabidopsis thaliana (Hong, Delauney & Verma, 2001), 12 CsCalSs in Citrus sinensis (Granato et al., 2019), 32 BnCalSs in Brassica napus (Liu, Zou & Fernando, 2018), eight VvCalSs in Vitis vinifera (Yu et al., 2016), seven HvCalSs in Hordeum vulgare (Schober et al., 2009), 15 BraCalSs in Chinese cabbage (Pu et al., 2019). Generally, according to the evolutionary analysis of CalSs in the above species, the CalS family can be divided into three of four main groups (Chen & Kim, 2009; Liu, Zou & Fernando, 2018).

Cotton is an important economic crop in China which yield is affected by biotic and abiotic stresses, producing prevalent fibers for textile industry (Wang et al., 2020; Fang et al., 2014). Some researches have reported that cotton fiber elongation was related to callose deposition which may be involved in the closing of plasmodesmata, then promoted the fiber length (Ruan et al., 2004). It is possible that callose affects fiber elongation by controlling CalS. In light of the above, the CalS may play a significant role in cotton in responsing various stresses and promoting fiber elongation. A callose synthase gene, CFL1 was identified (Cui et al., 2001), however, the callose synthase gene family members, phylogenetic relationships and expression patterns in cotton are still unclear.

In the current study, we identified callose synthase genes in two cultivated allotetraploids cotton, G. hirsutum and G. barbadense, and their two putative genome donors, G. raimondii and G. arboreum, then discussed their phylogenetic relationships, conserved domains, gene structures, synteny, and cis-acting elements. We also focused on the expression patterns of GhCalSs in various tissues and their expression under abiotic stress. These findings provide a solid foundation for further study of the roles of CalSs in cotton fiber development and stress responses.

Materials & Methods

Plant materials and treatments

Upland cotton TM-1, planted in Anyang Institute of Technology, was subjected to salt stress (350 mM NaCl) and drought stress (12% PEG6000) when the seedling reached two weeks. The leaves were collected 0 h, 1 h, 3 h, 6 h, 12 h, and 24 h after treatment. CCRI45 and MBI7747 were planted on farms managed by Cotton Research of Chinese Academy of Agricultural Sciences in Anyang. Cotton fibers were collected at 5, 10, 15, 20, 25 days post-anthesis (DPA). All samples were stored at −80 °C.

Identification of CalS family members in Gossypium spp

The genome sequences and annotated files of G. hirsutum (Hu et al., 2019), G. barbadense (Hu et al., 2019), G. raimondii (Paterson et al., 2012), and G. arboretum (Du et al., 2018) were downloaded from Cottongen (https://www.cottongen.org/) (Yu et al., 2014). Both blast and HMMER were used to identify the CalS sequences. The 1,3 beta-glucan synthase (PF02364) and FKS1_DOM1 domain (PF14288) from the Pfam database (http://pfam.xfam.org/) were searched by the HMMSearch program in TBtools to determine the presumed protein sequence (Chen et al., 2020). Besides, 12 Arabidopsis CalSs (Hong, Delauney & Verma, 2001) were used as queries sequences to identify family members using the Blastp program of TBtools (Chen et al., 2020). The protein sequences without above two domains were rejected and the domain which incomplete were also deleted. Finally, the final sequences were calculated by using ExPASy (https://www.expasy.org/) to calculate the theoretical isoelectric points (pI) and molecular weights (MW) and using the CELLO (http://cello.life.nctu.edu.tw/) for subcellular localization prediction (Yu et al., 2006).

Phylogenetic tree construction, gene structure, and motif analysis

The phylogenetic tree among four Gossypium species and Arabidopsis thaliana was constructed by MEGA7 (Kumar, Stecher & Tamura, 2016). It was constructed by the neighbor-joining (NJ) method, with 1,000 bootstrap replicates, then was drawn by using EvolView (He et al., 2016). TBtools was used to extract the location information of CalSs and visualize the gene structure. MEME (https://meme-suite.org/meme/tools/meme) was used to identify the conservative motif with the parameter set to the maximum number of motifs: 20.

Chromosome location and synteny analysis for CalSs

The locations of CalSs on chromosomes were shown by TBtools using four cotton species genomic annotation files (Chen et al., 2020). MCScanX was used to analyze the collinearity of the CalSs, that is, using CalS protein sequences to analyze the orthologous and paralogous gene pairs (Chen et al., 2020). Collinear gene pairs were visualized by using the circos (Chen et al., 2020). To investigate the selection pressure between homologous genes, we calculated the nonsynonymous substitutions rate (Ka) and synonymous substitutions rate (Ks) of homologous genes by KaKs_Calculator (Wang et al., 2010).

Analysis of Cis-acting element in promoters and functional enrichment analysis

The 2,000 bp sequence upstream of the translation initiation codon ATG of CalS gene was selected as promoter. The cis-acting elements contained in the promoter region of the CalSs were predicted using the PlantCare website (Lescot et al., 2002). For functional enrichment analysis, gene ontology (GO) analysis was performed using the OmicShare tool (https://www.omicshare.com/tools).

GhCalSs expression patterns under different tissues and abiotic stresses

In order to analyze the expression of GhCalSs in different tissues and under stress, we downloaded 11 tissues (bract, pental, torus, root, leaf, stem, pistil, sepal, anther, ovule, fiber) and abiotic stress treatment (cold, heat, drought, salt) data from Cotton Omics Database (http://cotton.zju.edu.cn) (accession number: PRJNA490626) (Hu et al., 2019). GhCalSs with FPKM > 1 were considered as expressed genes. The expression patterns of the GhCalSs were visualized by ComplexHeatmap (Gu, Eils & Schlesner, 2016) based on the value of log2(FPKM+1).

RNA isolation and qRT-PCR analysis

FastPure Plant Total RNA Isolation Kit (RC401, Vazyme) was used to extract RNA, and then we used 1µg to synthesize cDNA (HiScript III 1st Strand cDNA Synthesis Kit, R312 Vazyme). ChamQ Universal SYBR qRT-PCR Master Mix (Q711, Vazyme) was used for qRT-PCR in ABI 7500 Fast Real-time PCR System (Applied Biosystems, USA). Gene-specific primers for qRT-PCR were designed by using primer-blast in NCBI, with melting temperatures of 55–60 °C, product lengths of 101–221 bp, primer length of 18–25 bp (Table S1). For qRT-PCR, the reaction contains 10 µL 2x ChamQ Universal SYBR qPCR Master Mix, 0.4 µL of each primer, 3 µL template, and ddH2O to make up the total 20 µL volume. Then it was carried out in the following condition: one cycles of 95 °C for 30 s, 40 cycles of 95 °C for 10 s and 60 °C for 30 s. Each experiment was repeated three times, and two of the completed data were selected for drawing. Expression of all genes were calculated using a 2−ΔΔCt method (Livak & Schmittgen, 2001).

Results

Identification and characterization of CalSs in Gossypium spp

Through the analysis of the CalS protein sequences in Arabidopsis thaliana, we found that all the protein sequences contain 1, 3-β-glucan synthase (PF02364) and FKS1_DOM1 domain (PF14288), total of 27 members of the CalS gene family in G. hirsutum, 28 in G. barbadense, 15 in G. raimondii, and 16 in G. arboretum were identified, all of which were named according to their chromosomal locations. The properties of CalSs in cotton were further analyzed (Table S2). The protein sequence length of CalSs ranged from 1,494 to 1,979 amino acids, with an average MW of 212.88 kDa, and shared high similarity to the Arabidopsis thaliana CalS proteins (Hong, Delauney & Verma, 2001). The isoelectric point (PI) values of the above genes were all greater than 7, indicating that CalSs in cotton were alkaline, which was the same as the biochemical properties of the CalSs in Chinese cabbage and Brassica (Pu et al., 2019; Liu, Zou & Fernando, 2018). The CalSs were most likely localized in the plasma membrane, as predicted in Arabidopsis thaliana and Chinese cabbage (Pu et al., 2019; Zavaliev et al., 2011).

Classification and phylogenetic analysis of the cotton CalSs

In order to investigate the evolutionary relationships of the CalSs in the four cotton species and its relationship with Arabidopsis thaliana, a phylogenetic tree was constructed using the protein sequences of CalSs (Fig. 1). Based on the phylogenetic tree of this study, the CalSs were divided into five groups. The distribution of CalSs in each group was shown in Table S3. The members of Group A were homologous to AtCalS11/AtCalS12, the members of Group B were homologous to AtCalS9/AtCalS10, the members of Group C were homologous to AtCalS6/AtCalS7, the members of Group D were homologous to AtCalS8, and the members of Group E were homologous to AtCalS1-5. There were Arabidopsis genes homologous to cotton in each group, further indicating that the cotton CalSs and the Arabidopsis thaliana CalSs were close in evolutionary, which was consistent with the evolutionary relationship between Arabidopsis and cotton. It is observed that most of the CalSs derived from At-subgenome of two cultivated allotetraploids cotton stayed close together with the CalS gene of G. arboretum, and the CalSs of Dt-subgenome stayed close together with the CalS gene of G. raimondii, which was consistent with the hypothesis that the allotetraploid cotton species were produced by the recombination of two diploid cotton species (Liu et al., 2015). Phylogenetic tree analysis suggested that the CalS homologous gene in cotton may have similar functions.

Figure 1 Phylogenetic analysis of CalS protein from G. hirsutum, G. barbadense, G. raimondii, G. arboretum and Arabidopsis.

The bootstrap values are shown at the nodes. The CalSs from G. hirsutum, G. barbadense, G. raimondii, G. arboretum, and Arabidopsis are marked with red check, orange rect, purple star, green triangle, grey circle, respectively.

Gene structure and amino acid motif analysis of the CalSs

The diversity of gene structure and differences in conserved motifs are the manifestations of the evolution of multigene families (Magwanga et al., 2018). The distribution of exon/intron regions of CalSs was analyzed to understand the diversity of gene structure (Fig. 2). The number of CalSs exons varied from 1–51, and most CalSs had more than 35 exons (57/86, 66.2%). Clearly, these CalSs were divided into an exon-poor group (<7 exons, group A) and other exon-rich groups (>37 exons, group B–E) (Fig. 2, Table S2). The exons of CalSs had high similarity in the same group, and the number of exons in group B, group D, and group E were the same (Fig. 2, Table S2).

Figure 2 Conservative motif and exon-intron structure of CalS genes in cotton.

(A) The evolutionary tree of CalSs was constructed using MEGA7. (B) Conservative motif of CalSs. The 20 motifs are displayed in different colored boxes. (C) Exon-intron structure of CalSs. Introns are presented by grey lines, exons by green boxes, and UTR for yellow boxes.

The motif is a conserved region in the sequence (Magwanga et al., 2018). We identified 20 possible motifs using MEME (Fig. 2). Interestingly, all CalSs except GbCalS2/14/15 and GhCalS1 contained motif1-20 and were arranged in the sequence of motif15-9-8-13-12-11-7-20-14-16-3-2-1-6-19-5-17-4-18-10. The distribution of CalSs were slightly different among different groups, and only the number and arrangement position were different. The number and arrangement of motifs in the same group of CalSs were more similar than those in other groups.

Chromosomal location, gene duplication, and syntenic analysis of the CalSs in Gossypium spp

Based on the sequencing and annotated information of the four cotton genomes, the chromosome length and the distribution of genes on chromosome could be analyzed (Fig. 3). The distribution of CalSs in the two heterotetraploid cotton species chromosomes was highly similar. For example, CalSs had the same number and distribution on chromosomes A03, A04, A05, A08, A11. In G. hirsutum, 27 GhCalSs were distributed on 15 chromosomes, including 13 GhCalSs in At-subgenome and 14 GhCalSs in Dt-subgenome. In G. barbadense, 28 GbCalSs were distributed on 16 chromosomes, including 14 GbCalSs in At-subgenome and 14 GbCalSs in Dt-subgenome. In G. arboreum, 16 GaCalSs were distributed on eight chromosomes and a scaffold. In G. raimondii, 15 GrCalSs were distributed on eight chromosomes. Most CalSs occured at the upper or lower arms of chromosomes. D08 and D10 chromosomes both had the largest number of CalSs in the two allotetraploid cotton. Obviously, chromosome length was not positively correlated with the distribution number of CalSs on chromosomes.

Figure 3 Distribution of 86 CalSs on cotton chromosomes.

The chromosome name is on the left of each chromosome, and the gene ID is on the right. (A) G. hirsutum; (B) G. barbadense; (C) G. arboretum; (D) G. raimondii.

Gene duplication is the basis for the functional differentiation of homologous genes, the main reason for the generation of new functional genes (Conant & Wolfe, 2008). In order to explain the gene replication events of CalSs in cotton, we identified 15, 14 paralogous gene pairs in G. hirsutum, G. barbadense respectively, and one pair in G. arboreum. But there was no paralogous gene pair in G. raimondii (Table S4). GhCalS21/22, GbCalS1/2 as well as GbCalS22/23 were tandem duplication. In the four cotton species, the duplication events of the CalSs were WGD/Segmental, Tandem Duplicates, Dispersed, and proximal duplication, and the main expansion mechanism was WGD/Segmental (Table S2).

In order to illustrate the collinearity of CalS genes, we analyzed the orthologous and paralogous gene pairs (Fig. 4, Table S4). There were 31 CalS orthologous gene pairs among G. arboretum and two allotetraploid cotton species, including 15 pairs between with At-subgenome of G. hirsutum and 16 pairs between with At-subgenome of G. barbadense. There were 17 CalS orthologous gene pairs among G. raimondii and two allotetraploid cotton species, including nine pairs between with Dt-subgenome of G. hirsutum and eight pairs between with Dt-subgenome of G. barbadense. Meanwile, Ka/Ks of CalS homologous pairs were calculated to further understand the adaptation of the CDS region of CalSs (Fig. 4, Table S5). Most of the homologous gene pairs Ka/Ks < 1, and about 94.6% gene pairs had a Ka/Ks ratio less than 0.5, which meant that almost all gene pairs underwent purification selection. Only Ka/Ks > 1 of GaCalS2/GbCalS1 indicated that this was a positive selection for beneficial mutations.

Figure 4 Collinearity analysis of CalSs in tetraploid and diploid cotton.

(A) Orthologous and paralogous gene pairs among tetraploid and diploid cotton species. The lines represented by various colors indicates the syntenic regions around CalSs, and the color between the same species is the same (B) Ka, Ks, Ka/Ks distribution of CalS gene pairs. Ka, Ks, Ka/Ks analysis of GbCalS-GaCalS, GbCalS-GbCalS, GbCalS-GhCalS, GbCalS-GrCalS, GhCalS-GaCalS, GhCalS-GhCalS, GhCalS-GrCalS.

Analysis of Cis-acting elements in promoter

Transcription factors can be combined with cis-elements in the promoter region to regulate gene transcription. Investigation of upstream regulatory sequence can help us to well understand the regulation mechanism and also supportive to estimate the potential function of the gene (Fig. 5, Table S6). Given the effect of plant hormones in abiotic stress, we focused on plant hormone responsive elements in promoter regions. ABA- (ABRE), auxin- (AuxRR-core, TGA-element), Gibberellin- (GARE-motif, P-box, TATC-box), MeJA- (CGTCA-motif, TGACG-motif), SA- (TCA-element) responsive elements were found in the promoters of 18, 6, 14, 18, 12 GhCalSs. All GhCalSs contained hormone response elements except the GhCalSs in Group D and GhCalS5 in Group C. More than half of the GhCalSs contained ABA/GA/MeJA-responsive elements. Auxin-responsive elements only in GhCalS6/14/20/21/22/23. Meanwhile, we also paid attention to elements related to stress. Low-temperature- (LTR), wound- (WUN-motif), drought- (MBS), stress- (TC-rich repeats), anaerobic induced response element (ARE), anoxic specific inducibility element (GC-motif) were found in the promoters of 11, 2, 15, 9, 23, 4 GhCalSs. Wound-responsive elements only in GhCalS5 and GhCalS18. Anoxic specific inducibility elements only in GhCalS11/15/17/25. In addition, these results suggested that CalSs might regulated by hormone and abiotic stresses.

Figure 5 Cis-acting elements on promoters of the GhCalSs.

(A) The evolutionary tree of GhCalSs was constructed wsing MEGA7. (B) The cis-acting element on the promoter of GhCalSs. Number of each cis-acting element in the promoter region.

Expression patterns of the GhCalSs under abiotic stresses

Previous studies have reported that the CalSs respond to abiotic stresses (Cui & Lee, 2016; Jacobs et al., 2003; Fromm et al., 2013). To understand the response of GhCalSs, we used public RNA-seq data of TM-1 treated with cold, hot, NaCl, and PEG to observe the expression patterns of GhCalSs (Fig. 6). Interestingly, all expressed GhCalSs were induced by different abiotic stresses, and the expression patterns were different. The expression of GhCalS3 was significantly up-regulated under cold, hot, NaCl, and PEG. The expression patterns of GhCalSs in the same group were slightly consistent, such as GhCalS3 and GhCalS6, GhCalS2 and GhCalS16. In order to verify the results obtained by the above transcriptome, cotton seedlings were treated with PEG and NaCl, and then the GhCalS2/3/6/9/16 were selected for qRT-PCR (Fig. 7). The expression of GhCalS3 and GhCalS6 in Group A were up-regulated within 24 h under PEG treatment and reached the peak at 24 h. The expression of GhCalS2, GhCalS9, GhCalS16 were up-regulated at first and then down-regulated and last up-regulated after PEG treatment. After NaCl treatment, there was no consistent trend of gene expression. GhCalS3 was significantly induced by NaCl and significantly up-regulated at 3 h. Both GhCalS2 and GhCalS16 of Group B were down-regulated within 24 h. The expression of GhCalS6 and GhCalS9 reached a peak at 12 h. These findings indicated that the expression patterns of several GhCalSs were changed after treatment, which proved that GhCalSs increased adaptability to abiotic stress (Fig. 6).

Figure 6 RNA sequence profiling of the CalS gene family.

(A) Heatmap displaying expression of expressed GhCalSs under hot, cold, NaCl, and PEG treatment (B) Heatmap displaying expression of expressed GhCalSs in each tissue.

Figure 7 qRT-PCR results of GhCalSs under PEG and NaCl.

Enrichment analysis of the GhCalSs

In order to further understand the function of GhCalSs, we carried out functional enrichment annotation of gene ontology (GO) using pvalue of ≤ 0.05 as the cutoff. The results improved our accurate understanding of gene function, including many significantly enriched terms (Fig. 8, Table S7). The GO-BP enrichment results showed 34 terms such as (1->3)-beta-D-glucan biosynthetic process (GO:0006075), beta-glucan metabolic process (GO:0051273), cellular carbohydrate biosynthetic process (GO:0034637), cellular macromolecule biosynthetic process (GO:0034645). The GO-CC enrichment results discovered 16 terms such as 1,3-beta-D-glucan synthase complex (GO:0000148), plasma membrane protein complex (GO:0098797), transferase complex (GO:1990234), catalytic complex (GO:1902494). The CC terms enriched by GO were consistent with the subcellular localization of GhCalSs. GO-MF enrichment exposed 8 terms, including 1,3-beta-D-glucan synthase activity (GO:0003843), UDP-glucosyltransferase activity (GO:0035251), catalytic activity (GO:0003824), hexosyltransferase activity (GO:0016758). In short, the GO enrichment results confirmed the function of the GhCalSs in many biological processes, which were associated with 1,3-β-D-glucan synthetic activity, hydrolyzase activity, and membrane parts.

Figure 8 Bubble plot showing GO enrichment analysis of GhCalSs.

The top 20 GO terms significantly enriched by GhCalSs.

GhCalSs expression patterns in various tissues and their role in fiber development

We used transcriptome data from different tissues of GhCalSs to gain insight the tissue-specific expression patterns of cotton. For instance, GhCalSs were expressed in various tissues, and some of them were highly expressed. Some of GhCalSs were expressed in one or more tissues (GhCalS2, 3, 4, 6, 8, 9, 15, 16, 19, 20, 21). However, the expression of a few genes (GhCalS1, 5, 7, 10, 11, 12, 13, 14, 17, 18, 22, 23, 24, 25, 26, 27) did not show any expression in any tissues.

In order to determine the effect of GhCalSs in cotton fiber development, we focused on the expression of GhCalSs in different fiber developmental stages of two samples, MBI7747 and CCRI45, with different lengths and strengths (Lu et al., 2017) (Fig. 9). The expression of GhCalS4 was the highest in TM-1, MBI7747, CCRI45 fiber tissue, so it was speculated that GhCalS4 had an important function in cotton fiber development. In order to further determine its function in fiber development, qRT-PCR was used to analyze the GhCalS4 expression differences in two samples (Fig. 9). The results showed that the expression level of GhCalS4 in the two samples gradually increased from 5 DPA to 25 DPA, which was consistent with the transcriptome data of TM-1 used above. The expression of GhCalS4 in CCRI45 was higher than in MBI7747 at 5DPA, 10DPA, 15DPA but was significantly lower than that of MBI7747 at 25DPA (Lu et al., 2017). Thus, GhCalS4 may be involved in cotton fiber elongation.

Figure 9 Expression patterns of GhCalSs in cotton fiber.

(A) The expression of GhCalSs of MBI7747 and CCRI45 at different fiber developmental stages. (B) qRT-PCR results of GhCalS4 at different fiber developmental stages.

Discussion

Callose plays a vital role in plant growth, development, and resistance to various adverse factors (Piršelová & Matušíková, 2012). The gene family of callose synthase has been identified in a variety of plants. In this study, we identified the CalSs in G. hirsutum, G. barbadense, G. raimondii, and G. arboreum, aiming to understand the role of CalS family in the cotton development.

A total of 86 CalSs were identified in four cotton species. They were divided into five groups based on evolutionary relationships. we divided CalSs (AtCalS9-12 homologous) into two groups due to the large difference of the CDS number, and the other groups were the same as those in Arabidopsis thaliana. The CalSs number was 2:2:1:1 in group B/C/D, which was consistent with the evolutionary relationship among cotton species (Table S3). Compared with Arabidopsis thaliana, different percentages existed between subgroups. The percentages of Group A and Group E were significantly different from those of the corresponding CalSs in Arabidopsis thaliana, suggesting that these genes in cotton may have functional differences with homologous genes in Arabidopsis thaliana to a certain extent. These results will help to validate the function of cotton CalS homologous gene with Arabidopsis thaliana.

Tetraploid cotton species are formed by natural crossbreeding between G. raimondii and G. arboretum (Wendel et al., 2009). Thus, the four cotton species are closely related in evolution. In Fig. 4, the orthologous gene pairs of CalSs were all clustered in the same branch or group. Phylogenetic and orthologous genes of CalS further indicated that the results of this study were consistent with the evolutionary view.

A large number of hormone-responsive elements were identified on GhCalSs promoters which may be involved in the regulation of GhCalSs. Salicylic acid (SA) was an endogenous signal molecule in plants (Loake & Grant, 2007). In Arabidopsis thaliana, the expressions of AtCalS1/5/9/10/12 were up-regulated by exogenous SA. Abscisic acid (ABA) played an important part in coping with a variety of adverse factors, closely related to callose synthesis (Liu et al., 2017). During the dormancy of Populus tomentosa buds, short-day induced ABA biosynthesis, promoted the expression of PtCalS1, callose deposited at the plasmodesmata to form blockage, which prevented the growth signal molecules from entering the cell and kept the dormancy state of buds (Tylewicz et al., 2018). Jasmonic acid (JA) was also involved in callose regulation, and Methyl Jasmonate (MeJA) application promoted callose deposition in grape leaves. Inhibition of the expression of Cationic peroxidase 3 (OCP3), a negative regulator of the JA pathway, increased callose deposition (Repka, Fischerová & Šilhárová, 2004). In conclusion, ABA, JA, and SA were involved in the regulation of callose deposition. GhCalSs promoters with ABA, SA and JA response elements were highly likely to be regulated by them in cotton. However, how CalS gene is regulated by these hormones in the face of biotic-abiotic stress or growth and development is not known, which needs to be further studied.

Callose deposition is one of a series of coping strategies in plants to abiotic stress. Low temperature stimulation of maize leaves increased callose content and reduced transport of photosynthate in phloem (Wu et al., 2018). In Arabidopsis thaliana, AtCalS7, AtCalS8 and AtCalS12 were associated with callose synthesis under the condition of wound (Jacobs et al., 2003; Cui & Lee, 2016; Xie et al., 2011). In this study, public transcriptome data were used to analyze the responses of cotton leaves to cold, heat, salt and drought, and qRT-PCR was used to verify the results, which showed that CalSs were involved in abiotic stresses.

Callose deposits regulate material transport and control plant development. In this study, GhCalS4 was highly expressed in fibers and differentially expressed in MBI7747 and CCRI45 fibers at each fiber developmental stage (5/10/15/20/25 DPA). It has been reported that callose deposition may be involved in the closure of plasmodesmata, and the closure of plasmodesmata had an important function in the elongation of cotton fibers (Ruan et al., 2004). In Sea Island cotton, plasmodesmata remain open longer than in Upland cotton, allowing sucrose to be fed into fibroblasts, which eventually increase osmotic potential by hydrolysis to fructose and glucose. The more soluble sugar, K+ accumulated, the higher the cell leavening pressure, which promoted the elongation of cotton fiber (Hu et al., 2019). MBI7747 is a chromosome segment substitution line (CSSL) with different genetic background constructed by crosses between the upland cotton CCRI45 as the recurrent parent and the Sea Island cotton Hai 1 with outstanding fiber quality through the combination of high-generation backcrossing and molecular marker-assisted selection. The fiber length and strength of MBI7747 are better than CCRI45. In CCRI45, the expression level of this gene at 5DPA, 10DPA and 15DPA were all higher than those of MBI7747 during fiber elongation, and it was speculated that the degree of callose deposition in MBI7747 was lower than that of CCRI45, which made more sucrose input into fiber cells to increase osmotic potential and promote fiber elongation. Thus, GhCalS4 may be an introgression gene or there was difference in epigenetic regulation.

Conclusions

In this study, we identified 86 CalSs from G. hirsutum, G. barbadense, G. raimondii, and G. arboreum using conserved domains. Phylogeny, gene structure, motif, chromosome location and homologous genes were analyzed. It indicated that CalSs have been highly conserved during evolution by the analysis of CalSs structure, conversed motifs, and syntenic blocks. WGD/Segmental replication was the main driving force for the amplification of CalS family in cotton, and purification selection played an important role in the evolution of CalSs. In addition, the cis-acting elements of GhCalSs related to hormone regulation and development and their expression patterns in stresses and tissues were also analyzed. CalS gene can be induced by abiotic stress. Furthermore, the expression difference of GhCalS4 in fiber of different length and strength materials was analyzed. It was speculated that GhCalS4 played a major role in fiber elongation. These findings could lay a foundation for further study on the role of CalS gene in stress response and fiber development.

Supplemental Information

Supplemental Information 1 Primers used in this study

Click here for additional data file.

Supplemental Information 2 List of the identified CalS genes in cotton

Click here for additional data file.

Supplemental Information 3 List of the number of CalS family genes

Click here for additional data file.

Supplemental Information 4 Orthologous and paralogous CalS gene pairs among G. hirsutum, G. barbadense, G. raimondii, and G. arboretum.

Click here for additional data file.

Supplemental Information 5 Distribution of Ka, Ks, Ka/Ks of CalS gene pairs

Click here for additional data file.

Supplemental Information 6 Cis-acting elements in the promoter regions of GhCalSs.

Click here for additional data file.

Supplemental Information 7 The GO enrichment analysis of GhCalSs

Click here for additional data file.

Supplemental Information 8 Raw data: qRT-PCR results of GhCalSs under salt, drought, and different fiber development stages

Click here for additional data file.

Additional Information and Declarations

Competing Interests

Author Contributions

Data Availability

The authors declare there are no competing interests.

Jiajia Feng performed the experiments, analyzed the data, prepared figures and/or tables, authored or reviewed drafts of the paper, and approved the final draft.

Yi Chen and Xianghui Xiao performed the experiments, analyzed the data, prepared figures and/or tables, and approved the final draft.

Yunfang Qu and Pengtao Li analyzed the data, authored or reviewed drafts of the paper, and approved the final draft.

Quanwei Lu and Jinling Huang conceived and designed the experiments, authored or reviewed drafts of the paper, and approved the final draft.

The following information was supplied regarding data availability:

The raw data are available in the Supplemental File.

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
