# Peer review of "Genome-wide analysis of the CalS gene family in cotton reveals their potential roles in fiber development and responses to stress"

_PeerJ, doi:10.7717/peerj.12557_

## Round 0.1 · original submission · Major Revisions

Dear author:

Your manuscript entitled "Genome-wide Analysis of the CalS gene family in cotton reveals their potential roles in fiber development and responses to stress" which you submitted to PeerJ, has been reviewed. The reviewer comments are included at the bottom of this letter.

The reviews are in general favourable and suggest that it will be suitable for publication subject to major revisions. Please consider these suggestions, and I look forward to receiving your revision.

When you revise your manuscript please highlight the changes you make in the manuscript by using the track changes mode in MS Word or by using bold or coloured text.

Once again, thank you for submitting your manuscript to PeerJ and I look forward to receiving your revision.

Sincerely,

Xin Liu
Editor, PeerJ

·

Basic reporting

no comment

Experimental design

no comment

Validity of the findings

no comment

Additional comments

PeerJ MS#65037
The manuscript by Feng et al. describes the identification and analysis of callose synthase (CalS) genes in the 4 cotton genomes. They characterized 27, 28, 16 and 15 CalS from G. hirsutum, G. barbadense, G.raimondii and G. arboretum, respectively, by BLAST search the cotton genome. Phylogenetic analysis of these CalS genes in combination of those from Arabidopsis revealed 5 clades with distinct gene structures. Selection analysis indicate that these CalS proteins have undergone purify selection during cotton evolution. Further expression analysis and cis element annotation enable the authors to connect the expression CalS genes with hormonal and abiotic stress responses.
The manuscript is generally well-written and contains certain amount of work. The results were clearly described and presented.
I have a few minor points to improve the manuscript:
1. Line 14, “Callose synthase” describe a gene and should be italicised.
2. Line 23 and elsewhere in the MS, change GHCalSs into GhCalSs.
3. Line 48-50, please be aware the space between the parentheses and the text.
4. Line 59, change “significant” to “important”.
5. Line 59-66, I found myself had a problem in seeing the key question or aim of this study from this paragraph. I would suggest the authors to re-phrase this paragraph.
6. Line 67, “donors” to “genome donors”
7. Line 90, please explain why “incomplete sequences were deleted”
8. Line 110, please define the promote more accurately, “upstream the CalS gene is obscure”.
9. Line 120, in the section of “RNA isolation and qRT-PCR analysis”, please include all the details of how the experiments were conducted, for example, how much RNAs were used as templet for reverse transcription, were the specificity of qRT-PCR primers verified by PCR-Sequencing and Standard Culve, as such, the authors should be aware the transparency of the experiments.
10. Line 177, “chromosomes” to “chromosome”
11. Line 185, “upper and lower ends..” to “upper and lower arms..”
12. Line 207, I appreciated the authors’ efforts in mining the cis-elements in the regulatory regions of all the CalS genes. Nonetheless, a control gene/genes with related function should be set up for comparison.
13. Line 220-221, all these CalS gens at best can be categorized as “abiotic/hormonal responsive” genes based on the presence of corresponding cis-elements in the promoters but rather “abiotic/hormonal signaling”.
14. Line 224, “researches” into “studies”.
15. Line 221-222, the last sentence of this paragraph makes no sense from the data presented, I would suggest to delete it.
16. Line 231, “the above hypothesis”, which hypothesis?
17. Line 242-247, the information of this paragraph is somehow misleading, please re-phrase.
18. Line 255, “materials” into “samples”
19. Line 260, “part” into “role”
20. Line 262-263 and elsewhere in the text and figure legend, latin names should be italicised.
21. Line 280, “distributed” into “identified”
22. Line 286, “short-day induced ABA increased” into “short-day induced ABA biosynthesis/signaling”

Reviewer 2 ·

Basic reporting

non comment

Experimental design

no comment

Validity of the findings

Conclusions were made, however, the conclusion of "CalS gene family in cotton reveals their potential roles in fiber development and responses to stress validation“ lacks evidences from function aspects.

Additional comments

The paper presents the data analysis for the genome sequence of cotton, and the conclusion of "CalS gene family in cotton reveals their potential roles in fiber development and responses to stress validation“ were based on sequence analysis. The evidences from function aspects, such as Gene Ontology, KEGG pathway can help.

---

## Round 0.2 · Minor Revisions

Dear Dr. Feng,
Thank you for your submission to PeerJ.

The reviews are in general favourable and suggest that the work will be acceptable, subject to minor revisions. Please consider these suggestions, and I look forward to receiving your revision.

·

Basic reporting

As far as I can see, the authors have made a great efforts in revising the manuscript. All my concerns have been adequately addressed. I dont have further comments.

Experimental design

N/A

Validity of the findings

N/A

Additional comments

As far as I can see, the authors have made a great efforts in revising the manuscript. All my concerns have been adequately addressed. I dont have further comments.

Reviewer 2 ·

Basic reporting

no comment

Experimental design

Figure 8 is interesting. However, this figure should be revised. The gene numbers should be presented in this figure rather than only the significant degree. The significant level of each functional terms can be presented by the node size of each point. The GO bubble figure from clusterprofile package can help(OMICS. 2012 May; 16(5): 284–287).

Validity of the findings

no comment

Additional comments

no comment

---

## Round 0.3 · accepted · Accept

Dear Dr. Feng,

Thank you for your submission to PeerJ.

Callose deposition is important for plant growth and development, and stress reponses. The author used genome-wide analysis and qRT-PCR experiment to show GhCalSs were involved in abiotic stress. Additionally, bioinformatics showed CalSs were conserved during evolution and their cis-acting elements were related to hormone regulation and development. The research is meaningful to explore the role of GhCalSs in plant physiological metabolism.

According to the suggestions of our reviewers, the author has improved the paper. Therefore, we are pleased to accept your paper “Genome-wide Analysis of the CalS gene family in cotton reveals their potential roles in fiber development and responses to stress” in its current form.

Thank you for your contribution to PeerJ and we look forward to receiving further submissions from you.